# Enhancing Hepatocellular Carcinoma Surveillance: Comparative Evaluation of AFP, AFP-L3, DCP and Composite Models in a Biobank-Based Case-Control Study

**DOI:** 10.3390/cancers17142390

**Published:** 2025-07-18

**Authors:** Coskun O. Demirtas, Sehnaz Akin, Demet Yilmaz Karadag, Tuba Yilmaz, Ugur Ciftci, Javid Huseynov, Tugba Tolu Bulte, Yasemin Armutcuoglu Kaldirim, Feyza Dilber, Osman Cavit Ozdogan, Fatih Eren

**Affiliations:** 1Division of Gastroenterology and Hepatology, Marmara University School of Medicine, Istanbul 34854, Türkiye; tuba_yilmaz83@hotmail.com (T.Y.); ugurdr37@gmail.com (U.C.); cavidmurad1793@gmail.com (J.H.); tugbatolu139@gmail.com (T.T.B.); ykaldirim@yahoo.com (Y.A.K.); drfgunduz@yahoo.com (F.D.); osmanozdogan@yahoo.com (O.C.O.); 2Institute of Gastroenterology, Marmara University School of Medicine, Istanbul 34854, Türkiye; yilmz.demet@gmail.com (D.Y.K.); erenfatih78@yahoo.com (F.E.); 3Section of Digestive Diseases, Yale University, New Haven, CT 06520, USA; 4Department of Internal Medicine, School of Medicine, Marmara University, Istanbul 34854, Türkiye; sehnazakin96@gmail.com; 5Department of Medical Biology, School of Medicine, Marmara University, Istanbul 34854, Türkiye; 6Faculty of Medicine, Recep Tayyip Erdogan University, Rize 53200, Türkiye

**Keywords:** alpha-fetoprotein, lens agglutinin-reactive alpha-fetoprotein, des-gamma-carboxy prothrombin, hepatocellular carcinoma, GALAD, GAAP, ASAP

## Abstract

This study evaluated the diagnostic performance of various blood-based biomarkers and composite biomarkers models for detecting hepatocellular carcinoma (HCC) across different disease stages and liver disease etiologies. Using a referral biobank of 562 patients—including 120 healthy control participants, 277 with at risk chronic liver disease (CLD) and 165 patients with HCC—we compared the diagnostic performance of several models, including GALAD, GAAP, ASAP, Doylestown and aMAP, with biomarkers alone, including Alpha-fetoprotein (AFP), Lens culinaris agglutinin-reactive AFP (AFP-L3) and des-gamma-carboxy prothrombin (DCP). Their performance was assessed using newly identified optimal cut-offs, research-established thresholds and cut-offs targeting 90% specificity. Overall, the GALAD model showed the highest sensitivity, particularly in non-viral HCC cases. In patients with viral etiology, GAAP and ASAP showed comparable performance to GALAD. All three composite models significantly outperformed individual biomarkers and other biomarker–clinical models. These findings support the integration of these combined biomarker and clinical models into future personalized HCC surveillance strategies.

## 1. Introduction

Hepatocellular carcinoma (HCC), the predominant form of primary liver cancer, is the sixth most common cancer worldwide and the third leading cause of cancer-related deaths [1]. According to the Global Burden of Disease (GBD) 2021 study, there were approximately 529,202 incident HCC cases and 483,875 deaths related to liver cancer, representing increases of 25% and 26%, respectively, between 2010 and 2021 [2]. In Turkey, an estimated 5039 new HCC cases and 4929 related deaths were reported in 2022, reflecting a notable national burden in line with global trends [3]. These findings underscore the importance of continued efforts in HCC prevention, early detection, and effective management strategies. Current surveillance guidelines recommend screening populations with an annual HCC incidence of ≥0.2%, as early-stage detection markedly improves prognosis [4,5,6]. Despite advances in understanding risk factors and hepatocarcinogenesis, most HCCs are diagnosed at advanced stages, limiting therapeutic options and survival rates. Early detection remains the cornerstone for improving patient outcomes, yet current surveillance strategies face significant limitations. Biannual abdominal ultrasound (USG), with or without alpha-fetoprotein (AFP) testing, remains the standard method, but its sensitivity is compromised by operator dependency and patient-related factors such as obesity and bowel gas [7,8]. Although cross-sectional imaging modalities like computed tomography (CT) and magnetic resonance imaging (MRI) offer improved sensitivity, their widespread use is limited by cost, accessibility, radiation exposure, and logistical challenges [9,10].

Serum biomarkers constitute an attractive alternative for HCC surveillance due to their non-invasive, reproducible, and objective nature. Among these, AFP is the most extensively studied biomarker; however, its utility is limited by suboptimal sensitivity, particularly for early-stage tumors, as a significant proportion of early HCCs exhibit AFP-negative profiles [11,12]. Additional biomarkers, including Lens culinaris agglutinin-reactive AFP (AFP-L3), an HCC-specific glycoform of AFP, and des-gamma-carboxy prothrombin (DCP), also known as Protein Induced by Vitamin K Absence or Antagonist-II [PIVKA-II], have been proposed to improve detection. Several cohort studies demonstrated that AFP-L3 provides high specificity (~95%) at a 10% cut-off but is limited by low sensitivity (~50%), while DCP shows moderate sensitivity (~60%) and high specificity (~90%) [11,13,14,15]. However, when used individually, these markers do not sufficiently overcome diagnostic challenges associated with early detection of HCC, as is the case in AFP.

Given the heterogeneity of HCC in terms of etiology and tumor biology, composite models integrating biomarkers with clinical parameters have been developed to enhance diagnostic performance. Among these, the GALAD score, incorporating age, sex, AFP, AFP-L3, and DCP, has been extensively validated and demonstrated superior sensitivity and specificity across diverse populations and etiologies [16,17,18]. Similarly, GAAP and ASAP models have been proposed and externally validated, using the AFP, DCP, age and gender data, excluding the usage of AFP-L3, aiming to provide comparable performance with fewer biomarkers [19,20]. Other models have also been proposed, without using AFP-L3 and DCP, to improve HCC surveillance performance. For example, aMAP score, developed through an international multicohort collaboration, is based on age, gender, bilirubin, albumin, and platelets, and has been validated as a reliable risk stratification tool for guided surveillance strategies [21]. Another validated model is the Doylestown algorithm, incorporating serum levels of AFP, alanine aminotransferase (ALT), and alkaline phosphatase (ALP) with age and gender; it has been shown to improve HCC detection by 2–20% compared to AFP alone [22].

Despite these advancements, external validation studies in diverse clinical settings and patient populations remain necessary before widespread clinical implementation. Comparative evaluations of these models within the same cohort remain limited. Data from different populations are especially scarce, despite significant geographic variations in HCC etiology and surveillance practices. Therefore, we aimed to evaluate the diagnostic performance of individual biomarkers (AFP, AFP-L3, and DCP) and composite scoring models (GALAD, GAAP, ASAP, aMAP, and Doylestown) in a well-characterized cohort of patients with chronic liver disease (CLD) at risk of HCC. We assessed each biomarker and model across different threshold strategies, including optimal, established, and high-specificity cut-offs, and performed subgroup analyses based on HCC etiology and AFP status.

## 2. Materials and Methods

### 2.1. Study Design and Patient Selection

This was a case–control biobank study conducted at Marmara University, Institute of Gastroenterology, from January 2019 to January 2024. The study cohort comprised 562 adult participants, including 120 healthy control participants, 277 patients with CLD at risk of HCC, and 165 patients with HCC. All participants were enrolled from the Marmara University HCC Biobank, which includes prospectively collected serum samples and clinical data from patients with liver diseases. All serum samples and clinical data were retrieved from the biobank. Although the main focus of this study was the comparison between HCC and CLD groups, a healthy control group was included to establish baseline levels of biomarkers and composite models in a healthy population.

Inclusion criteria for the HCC group included a confirmed diagnosis of HCC based on radiological and/or histological criteria, in accordance with the European Association for the Study of the Liver (EASL) guidelines [4]. For the CLD group, inclusion required a confirmed diagnosis of cirrhosis or chronic hepatitis B, representing individuals at risk for HCC and recommended for surveillance. Exclusion criteria were a prior diagnosis of HCC, the presence of any extrahepatic malignancy, HCC-suspected lesions in the CLD group, and a history of liver transplantation. No subjects included in the study were receiving medications known to affect AFP, AFP-L3, or DCP levels—such as vitamin K antagonists or active HCC-specific therapies—to avoid confounding effects on biomarker measurements.

### 2.2. Demographic and Clinical Data Collection

Demographic characteristics, including age, gender, height, weight, and body mass index (BMI), were recorded for all participants. Comorbidities, including diabetes mellitus, hypertension, and hyperlipidemia, were assessed based on clinical histories and laboratory results. Disease etiology (hepatitis B virus [HBV], hepatitis C virus [HCV], metabolic-associated steatotic liver disease [MASLD], or alcohol-related liver disease [ALD], etc.) was noted. Laboratory parameters, including albumin, bilirubin, sodium, and international normalized ratio (INR), were recorded for all participants. For HCC patients, tumor characteristics and staging according to the Barcelona Clinic Liver Cancer (BCLC) system were recorded [23]. BCLC stages 0 or A were classified as early-stage HCC.

### 2.3. Serum Biomarker Measurements and Calculation of Scoring Models

Serum samples were collected from each participant, stored at –80 °C, and subsequently analyzed for the concentration of AFP, AFP-L3, and DCP using an automated immunoassay system assay on the μTASWako i30 immuno-analyzer (Wako Chemicals, Neuss, Germany) [24]. Analytical assay sensitivities were 0.3 ng/mL for AFP and 0.1 ng/mL for DCP. AFP-L3 was reported as a percentage of total AFP when AFP-L1 and AFP-L3 were 0.3 ng/mL or more. All assays were performed on the same sample and were performed in an external laboratory blinded to the clinical data.

The GALAD, GAAP, ASAP, aMAP and Doylestown models were calculated for each participant using their original formulas, as follows:

GALAD formula [25]: −10.08 + 1.67 × [Sex (1 for male, 0 for female)] + 0.09 × [Age] + 0.04 × [AFP−L3%] + 2.34 × log_10_[AFP] + 1.33 × log_10_[DCP]

GAAP formula [19]: −11.203 + 0.699 × [Sex (1 for male, 0 for female)] + 0.094 × [Age] + 1.076 × log_10_[AFP] + 2.376 × log_10_[DCP]

ASAP formula [20]: −7.57711770 + 0.04666357 × [Age] − 0.57611693 × [Sex (1 for female, 0 for male)] + 0.42243533 × ln[AFP] + 1.10518910 × ln[DCP]

aMAP formula [21]: ((0.06 × age + 0.89 × sex (male: 1, female: 0) + 0.48 × ((log10 total bilirubin × 0.66) + (albumin × −0.085)) − 0.01 × platelets) + 7.4)/14.77 × 100

Doylestown algorithm [22]: 1/(1 + EXP (− (−10.307 + (0.097 × Age) + (1.645 × Gender) + (2.314 × logAFP) + (0.011 × ALP) + (−0.008 × ALT))))

### 2.4. Statistical Analysis

Descriptive statistics were used to summarize the baseline characteristics. All group comparisons and diagnostic performance analyses were conducted using appropriate non-parametric inferential tests to ensure robust and valid statistical inference. Differences between groups were assessed using the chi-square test of Fisher’s exact test for categorical variables where appropriate, based on expected cell counts; and the Mann–Whitney U test for comparisons between two groups, and the Kruskal–Wallis test for comparisons among three groups for continuous variables.

To evaluate predictors of HCC occurrence, univariate binary logistic regression analyses were initially conducted for all variables including each biomarker and composite model. Variables demonstrating statistical significance in univariate analyses were subsequently included in multiple multivariable logistic regression models to identify independent association with HCC. To prevent multicollinearity, each multivariable model incorporated only one biomarker or composite model, alongside covariates that did not overlap with components of the included biomarker or model. Adjusted odds ratios with 95% confidence intervals, beta-coefficients, and *p*-values were reported for all regression models.

Diagnostic performance metrics, including sensitivity, specificity, positive predictive value (PPV), and negative predictive value (NPV), were calculated for each biomarker and scoring model. Receiver operating characteristic (ROC) curve analysis was generated to determine the area under the curve (AUC) for each biomarker and model in detecting any-stage and early-stage HCC, as well as AFP-negative HCCs defined as having AFP value below 20 ng/mL. Subgroup analyses were conducted, categorizing HCCs based on the etiology as viral and non-viral.

Diagnostic performance was evaluated by calculating the sensitivity, specificity, PPV and NPV for each biomarker and model, using different cut-off strategies:(1)Optimal cut-offs: optimal thresholds for each biomarker and scoring model were determined using the Youden index, which maximizes the sum of sensitivity and specificity.(2)Established cut-offs: performance was also assessed using previously validated cut-offs from the literature, allowing for comparison with established thresholds for clinical use.(3)90% specificity cut-off: to evaluate the ability of biomarkers and models to maintain high sensitivity, performance was assessed at cut-offs where specificity was strictly set at 90%.

Statistical significance was set at a *p*-value < 0.05, and all analyses were performed using the SPSS software version 27.0 (IBM, Armonk, NY, USA), R Open-Source Software version 4.0.3, and the MedCalc software version 9.2.1.0 (MedCalc, Ostend, Belgium).

### 2.5. Ethical Statement

This study was conducted in compliance with the Declaration of Helsinki and its subsequent amendments. Ethical approval was obtained under Ethics Committee Protocol Number 28.06.2024.768, dated 28.06.2024. Written informed consent was obtained from all participants at the time of Biobank sample collection, under Ethical approval (Ethics Committee Protocol Number 09.2019.716, dated 13.09.2019). Data used in this retrospective analysis were accessed for research purposes starting from 08.07.2024. During data collection and analysis, the authors had access to information that could identify individual participants; however, all data were anonymized prior to analysis to ensure confidentiality.

## 3. Results

### 3.1. Baseline Characteritics

A total of 562 individuals were included: 120 healthy controls, 277 CLD patients and 165 HCC patients. The median age significantly differed across groups (*p* < 0.001), with HCC patients being oldest (median 65 years), followed by CLD patients (median 54 years), and healthy controls (median 45 years). Male predominance was observed in the HCC group (77%) compared to CLD (59.6%) and healthy controls (53.3%) (*p* < 0.001). Diabetes mellitus (37.2% vs. 27.2%, *p* = 0.033) and hypertension (36.2% vs. 22.2%, *p* = 0.002) were more prevalent among HCC patients. The underlying etiology of liver disease differed among CLD and HCC groups, with HBV being the predominant cause in both, although significantly more frequent in the CLD group (68.2% vs. 53.3%, *p* = 0.002). Cirrhosis was markedly more common in the HCC group (90.9% vs. 52.7%, *p* < 0.001). CTP, MELD and ALBI scores were all significantly higher in HCC patients (all *p* values < 0.001). HCC patients also exhibited higher rates of ascites (*p* < 0.001), esophageal varices (*p* = 0.007), and variceal bleeding (*p* = 0.022). Laboratory parameters showed significantly lower albumin, higher total bilirubin, lower sodium, and higher INR in HCC patients compared to CLD patients (all *p*-values <0.001). Demographic, clinical, and laboratory characteristics of the study population are summarized in Table 1.

The median tumor size among HCC patients was 43.0 mm (IQR: 9–162 mm), with 57.3% presenting with a single lesion. Evidence of lymph node involvement, portal vein thrombosis (PVT), malignant vascular invasion and extrahepatic metastases was present in 15.9%, 18.3%, 11.6%, and 4.3% of HCC patients, respectively. According to BCLC staging system, most patients were diagnosed at intermediate-B (40.9%) or early-A (32.9%) stage. Overall, 38% of HCC cases were diagnosed at a resectable early stage, while the rest of the HCCs were at unresectable stages, detected beyond eligibility for curative treatment options. Tumoral characteristics are detailed in Appendix A.

### 3.2. Biomarkers and Composite Model Scores

Serum levels of AFP, AFP-L3, and DCP, as well as composite scores (GALAD, GAAP, ASAP, aMAP, Doylestown), differed significantly among the three groups (*p* < 0.001 for all comparisons). AFP was highest in HCC (median 16.8 ng/mL), compared to CLD (median 2.9 ng/mL) and healthy controls (median 2.1 ng/mL). AFP-L3 and DCP also showed marked elevation in HCC (15.9% and 4.8 ng/mL, respectively), while levels in CLD and healthy controls were generally low. All composite models had significantly higher scores in HCC compared to lower scores in CLD and healthy controls. Biomarker and composite model characteristics are also presented in Table 1.

In pairwise comparisons, all biomarkers and composite models were significantly higher in HCC versus CLD (*p* < 0.001 for all) (Appendix A). Most markers were also significantly higher in CLD in comparison to healthy controls, indicating some elevation in CLD even without malignancy. DCP was the exception, showing no significant difference between CLD and healthy controls (*p* = 0.337), but still markedly elevated in HCC. All individual tested biomarkers (AFP, AFP-L3, and DCP) and scoring models were significantly elevated among early-stage and any-stage HCC patients in comparison to CLD patients (all *p*-values <0.001) (Figure 1).

### 3.3. Logistic Regression Analyses

To assess the independent association of each biomarker and composite model with HCC occurrence, we conducted univariate and multivariate binary logistic regression analyses. In univariate analyses, age, gender, presence of diabetes mellitus, hypertension, cirrhosis, esophageal varices, history of variceal bleeding, and CTP, as well as all investigated biomarkers and models, were all significantly associated with HCC (all *p* < 0.05) (Appendix A). In the multivariate logistic regression models, the outcome variable was the presence of HCC. The models included 165 HCC cases (events) and 276 CLD controls without HCC. Multivariate logistic regression models showed that AFP (aOR:1.040, 95% CI: 1.016–1.064, *p* < 0.001), AFP-L3 (aOR:1.071, 95% CI: 1.051–1.092, *p* < 0.001) and DCP (aOR:1.015, 95% CI: 1.003–1.027, *p* = 0.011), as well as GALAD (aOR: 2.166, 95% CI: 1.769–2.651, *p* < 0.001), GAAP (aOR: 2.157, 95% CI: 1.759–2.646, *p* < 0.001), ASAP (aOR: 2.072, 95% CI: 1.704–2.519, *p* < 0.001) and Doylestown (aOR: 130.81, 95% CI: 37.257–459.277, *p* < 0.001) models, demonstrated strong independent associations with HCC (Table 2). The aMAP score, however, was not statistically significant in multivariable analysis (aOR:0.977, 95% CI: 0.935–1.021, *p* = 0.298).

### 3.4. Diagnostic Performance of Biomakers and Scoring Models

To evaluate the performance of biomarkers and scoring models to detect HCC, we used three different approaches: optimal cut-offs based on the Youden index, established cut-offs from the literature, and cut-offs ensuring 90% specificity.

#### 3.4.1. Performance Based on Optimal Cut-Offs

At optimal thresholds determined by Youden’s index, the GALAD model demonstrated the highest sensitivity for detecting any-stage HCC (90.3%), followed by GAAP (88.8%) and ASAP (87.5%), with specificities ranging from 80 to 85% (Table 3). Among individual biomarkers, DCP had the highest sensitivity (85.5%), while AFP had the highest specificity (84.8%) in detecting any-stage HCC. In early-stage HCC subgroup, GALAD again showed the highest sensitivity (89.1%), followed by GAAP (85.7%) and ASAP (84.1%), with specificities ranging from 73 to 76%. In AFP-negative HCC cases, GAAP had the highest sensitivity (84.7%), followed by GALAD (80.7%) and ASAP (77.4%), with specificities around 85%. GALAD also had the highest sensitivity in both viral (92.7%) and non-viral (72.7%) HCCs, with a specificity of 88%. GALAD, ASAP, and GAAP all achieved NPV above 90% for in all groups, except for non-viral HCC detection, where NPVs ranged from 75 to 81%.

#### 3.4.2. Performance Used in Established Cut-Offs

When applying previously validated cut-offs, GALAD retained the highest sensitivity for detecting any-stage (75.8%) and early-stage (57.8%) HCC, significantly outperforming the single biomarkers and other models, while maintaining high specificity (93.5%) (Table 4). GAAP and ASAP showed lower sensitivities but comparable high specificities. Notably, aMAP achieved good sensitivity (79.4%) for early-stage HCC but with very low specificity (65.9%), making it less suitable for screening. Using the −0.63 established GALAD cut-off, the model outperformed others by providing high sensitivity and NPV across all subgroups, without significant loss of specificity.

#### 3.4.3. Performance at the 90% Specificity Threshold

When adjusting thresholds to achieve 90% specificity, GALAD retained the highest sensitivity for detecting any-stage HCC (81.2%) (Table 5). In early-stage HCC, GALAD’s sensitivity dropped to 68.8% but remained the highest among models. Among individual biomarkers, DCP had the highest sensitivity for any-stage (74.6%) and early-stage (59.4%) HCC. For AFP-negative and viral HCCs, ASAP demonstrated increased sensitivity (70.6% and 92.6%, respectively), whereas for non-viral HCCs, GALAD maintained the highest.

### 3.5. Receiver Operating Characteristic (ROC) Curve Analysis

ROC curve analysis supported these findings, demonstrating that GALAD, ASAP and GAAP had the highest and nearly identical AUCs for detecting any-stage (0.931, 0.932 and 0.930, respectively) and early-stage (0.877, 0.879 and 0.879, respectively) HCCs, outperforming other models and individual biomarkers (Figure 2). In AFP-negative HCCs, GAAP showed slightly higher AUC (0.908 [0.874–0.943]), compared to GALAD (0.888 [0.847–0.928]) and ASAP (0.896 [0.858–0.896]). In the etiology-based subgroup analysis, GALAD, ASAP, and GAAP had similarly high AUCs (0.955, 0.960, and 0.958, respectively) for detecting viral HCCs (Figure 3). In non-viral HCCs, GALAD had the highest AUC at 0.872, outperforming other models and individual biomarkers.

## 4. Discussion

In this biobank-based case–control study, we comprehensively evaluated the diagnostic performances of individual serum biomarkers (AFP, AFP-L3, DCP) and composite scoring models (GALAD, GAAP, ASAP, aMAP, and Doylestown) for detecting HCC in different clinical settings. Consistent with previous literature, our findings demonstrate that composite models, particularly GALAD, GAAP and ASAP, significantly outperform individual biomarkers across various clinical settings, including early-stage and AFP-negative HCCs. These results reinforce the well-documented limitations of AFP as a standalone biomarker. Despite its high specificity (98.2% at the established cut-off and 84.8% at optimal cut-off), AFP exhibited only low to modest sensitivity (46.7% at established cut-off and 72.1% at optimal cut-off), especially in early-stage disease (34.4% at established cut-off and 64.1% at optimal cut-off). Both AFP-L3 and DCP improved sensitivity over AFP alone, in any-stage (both >80%) and early-stage HCC (both >70%) detection at optimal cut-offs. Using established cut-offs, AFP-L3 improved sensitivity by approximately 15% compared to AFP, whereas DCP showed lower sensitivity in that setting. To optimize their utility in screening, we analyzed biomarker performance at a fixed specificity of 90%, aiming to reduce false positives and false negatives, thereby avoiding unnecessary testing and associated healthcare burden. Using fixed 90% specificity cut-offs, DCP improved sensitivity by 11% for both any-stage and early-stage HCC compared to AFP alone, while AFP-L3 provided a declined sensitivity. In AFP-negative HCC, DCP demonstrated a more than 20% higher sensitivity than AFP-L3, demonstrating the superiority of DCP as an individual biomarker. Still, both AFP-L3 and DCP remained suboptimal as standalone tools, consistent with meta-analyses reporting sensitivity ranges of 34–64% for early- and any-stage HCC detection [14,26,27]. Notably, DCP showed superior sensitivity among the individual biomarkers in our study, reflecting its ability to detect tumors with minimal AFP secretion.

Composite models consisting of at least two biomarkers demonstrated superior diagnostic performance. The GALAD, GAAP, and ASAP scores consistently achieved the highest AUCs across all evaluated subgroups. GALAD had the highest sensitivity for detecting both any-stage and early-stage HCC, without a significant trade-off in specificity or NPV at either optimal or established cut-offs. Importantly, GALAD maintained strong performance even at 90% specificity, highlighting its suitability for real-world clinical settings where minimizing false positives is essential. These findings are in line with previous studies reporting GALAD AUCs >90% in diverse populations, including those with viral hepatitis and MASLD, supporting its diagnostic utility across global cohorts [28,29,30,31,32,33]. The most widely adopted GALAD cut-off value of −0.63, initially proposed in a MASH cohort, yielded sensitivity of 85.6% and specificity of 93.3% [34]. However, subsequent studies reported variable performance. A large multicenter Latin American study reported an AUC of 0.76 (sensitivity 70%, specificity 83%), and a European cohort found an AUC of 0.69 (sensitivity 66%, specificity 72%) using the −0.63 cut-off [35]. In our study, the −0.63 cut-off performed better in non-viral etiology, highlighting etiology-specific performance differences. In a US-based phase III biomarker trial, the −0.63 cut-off improved HCC detection, in comparison to AFP alone, but also increased false positives [36]. A subsequent phase III trial identified −1.36 as the optimal GALAD cut-off, yielding 62% sensitivity and 82% specificity [37]. A recent Asian prospective study determined a cut-off of −1.95, almost identical to our study’s optimal cut-off, which provided 75% sensitivity and 92.5% specificity [38].

Our findings underscore the consistency of GALAD score performance across different populations and its superiority over single biomarkers. While no globally standardized cut-off exists, our findings suggest −1.96 as the optimal cut-off for HCC detection in our population, with lower thresholds needed for early-stage disease. However, an optimal cut-off merely offers the best balance of sensitivity and specificity. A practical alternative may be to set specificity at 90%, as we did, to reduce false positives. In that scenario, the ideal GALAD cut-off for detecting both any-stage and early-stage HCC was −1.22. Given the differing biological behavior of viral and non-viral HCC, tailoring cut-offs based on etiology may enhance performance when using GALAD as a surveillance tool.

GAAP and ASAP scores, which exclude AFP-L3, performed nearly identical to GALAD, underscoring their non-inferiority. These models, developed to provide cost-effective alternatives by omitting AFP-L3, showed comparable AUCs across all subgroups, including early-stage, AFP-negative, and both viral and non-viral HCC. This observation suggests that simplified models may be preferable in resource-limited settings. AFP-negative HCC presents a particular diagnostic challenge, as it often evades standard surveillance. In our study, GAAP and ASAP outperformed individual biomarkers in AFP-negative cases, demonstrating superior sensitivity and NPV across all cut-off strategies. These results underscore the advantage of integrating demographic and clinical data to better reflect the heterogeneity of tumor biology.

Etiology-specific analyses further highlighted the strengths of these models. ASAP had slightly higher sensitivity than GALAD and GAAP in viral HCC, while GALAD remained superior in non-viral etiology, consistent with prior findings suggesting that biomarker expression varies by etiology [39]. Among non-biomarker-based models, the aMAP score demonstrated a high sensitivity for early-stage HCC at its established cut-off, but very low specificity (65%), which could lead to a high false positive rate, making it more suitable for risk stratification rather than a screening tool. The Doylestown algorithm improved performance over AFP alone but fell short of biomarker-based models, as observed in prior validation.

Our study has several strengths, including the use of prospectively collected biobank data, a clinically diverse real-world cohort, and a comprehensive evaluation of both individual biomarkers and composite models, with or without using a combination of biomarker combinations, using three different cut-off approaches. To the best of our knowledge, this is the most inclusive comparison study of proposed HCC screening models. Despite these strengths, several limitations should be acknowledged. First, the case–control design, though useful for biomarker analysis, may overestimate diagnostic accuracy compared to prospective surveillance studies. Second, the study was conducted using the biobank data from a referral center in Türkiye, which may limit the generalizability of our findings. However, this can also be viewed as a strength, as it represents the first study to provide data on these biomarker-based models from the Eastern European and Middle Eastern regions. Thirdly, our study did not include other emerging biomarkers such as Glypican-3 and osteopontin, which is a limitation worth noting. These biomarkers may offer additional diagnostic value; however, our focus was to compare established composite models and the core biomarkers they rely on, rather than to perform a broad evaluation of all potential serologic biomarkers for HCC. Finally, timely ultrasound data were not available for comparison, which could have further enriched the study.

## 5. Conclusions

Our results demonstrate the consistent performance of the GALAD score across diverse populations and underscore its superiority over individual biomarkers and other composite models. Notably, the GAAP and ASAP scores—which use one fewer biomarker (AFP-L3)—exhibited comparable performance, particularly in viral etiology. These results suggest that GAAP and ASAP may serve as cost-effective alternatives in resource constrained settings or where viral hepatitis dominates. In contrast, reliance on single biomarkers, as well as the use of aMAP and the Doylestown algorithm, is associated with lower diagnostic accuracy and may compromise screening effectiveness in clinical practice. Based on these findings, the GALAD score can be recommended as the current preferred screening tool for HCC, while the prospective validation of both established and optimized cut-off values in larger, multiethnic, real-world cohorts is warranted.

## Figures and Tables

**Figure 1 cancers-17-02390-f001:**
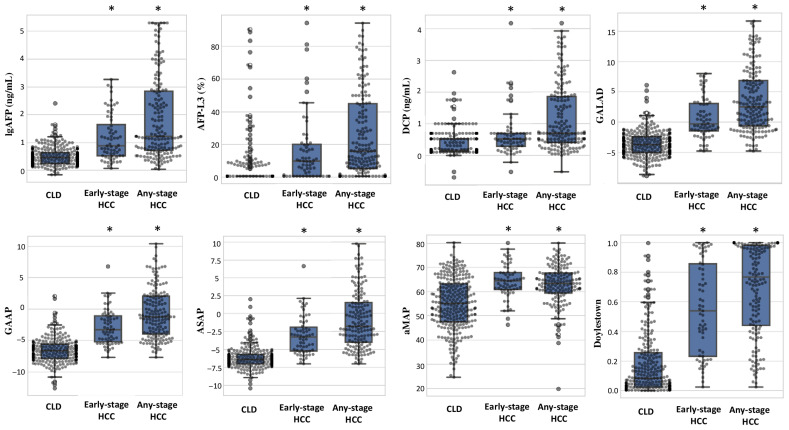
Comparison of individual biomarkers and composite models among patients with chronic liver disease, early-stage and any-stage hepatocellular carcinoma. CLD: chronic liver disease, HCC: hepatocellular carcinoma. * indicates *p* values <0.05 in comparison to reference patients with chronic liver disease (CLD).

**Figure 2 cancers-17-02390-f002:**
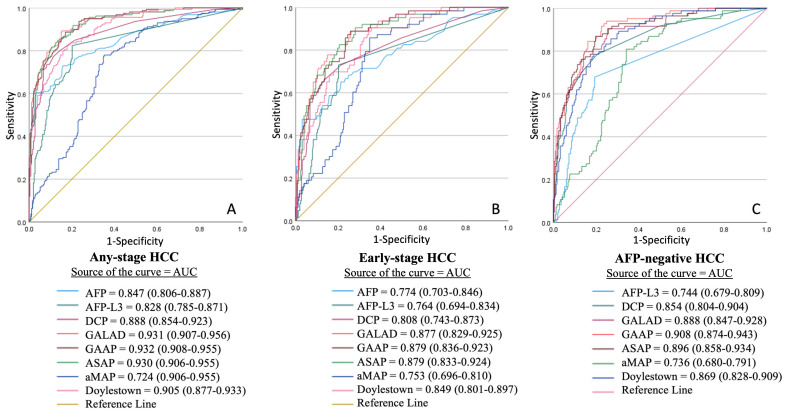
Receiver operating characteristic (ROC) analysis of the discriminatory performance of individual biomarkers and composite models for distinguishing (**A**) any-stage HCC, (**B**) early-stage HCC, and (**C**) AFP-negative HCC.

**Figure 3 cancers-17-02390-f003:**
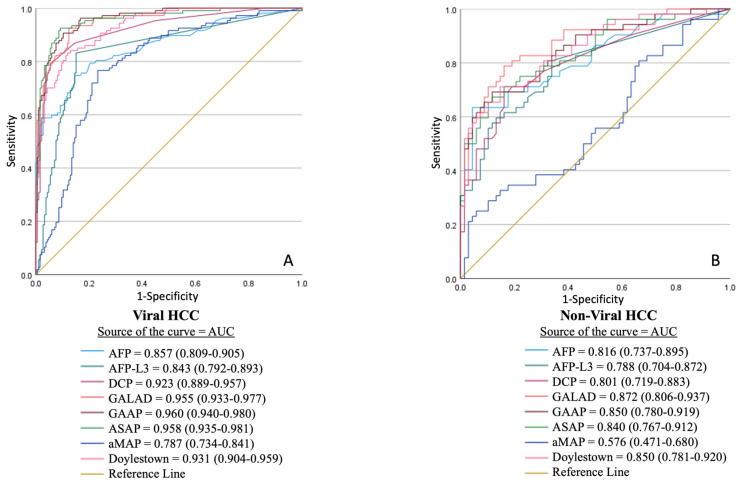
Receiver operating characteristic (ROC) analysis of the discriminatory performance of individual biomarkers and composite models for distinguishing (**A**) viral HCC and (**B**) non-viral etiology HCC.

**Table 1 cancers-17-02390-t001:** Baseline demographic, clinical, laboratory, and biomarker characteristics of the study population.

	Overall Cohort (*n* = 562)	Controls (*n* = 120)	CLD (*n* = 277)	HCC (*n* = 165)	*p*-Value
Age, median (IQR), years	59 (19–88)	45 (18–89)	54 (19–81)	65 (37–88)	<0.001 *
Gender, n (%)					<0.001 *
- Male	292 (66.1)	64 (53.3)	165 (59.6)	127 (77.0)	
- Female	150 (33.9)	56 (46.7)	112 (40.4)	38 (23.0)	
Body mass index, median (IQR),kg/m^2^	27.7 (16.5–46.7)	24.7 (18.9–29.7)	27.7 (16.9–46.7)	26.9 (16.5–45.7)	0.284
Diabetes mellitus, n (%)	136 (30.8)	-	75 (27.2)	61 (37.2)	0.033 *
Hypertension, n (%)	120 (27.1)	-	61 (22.2)	59 (36.2)	0.002 *
Hyperlipidemia, n (%)	36 (8.1)	-	18 (6.6)	18 (11.5)	0.103
Etiology, n (%)					0.002 *
- HBV	277 (62.7)	-	189 (68.2)	88 (53.3)	
- MASLD-cryptogenic	105 (23.8)	-	57 (20.6)	48 (29.1)	
- HCV	34 (7.7)	-	12 (4.3)	22 (13.3)	
- ALD	14 (3.2)	-	10 (3.6)	4 (2.4)	
- Autoimmune—PBC	10 (2.3)	-	7 (2.5)	3 (1.8)	
- Wilson’s disease	1 (0.2)	-	1 (0.4)	-	
- Budd–Chiari	1 (0.2)	-	1 (0.4)	-	
Cirrhosis, n (%)	296 (67.0)	-	146 (52.7)	150 (90.9)	<0.001 *
CTP score, median (IQR)	5 (5–12)	-	5 (5–10)	6 (5–12)	<0.001 *
MELD score, median (IQR)	8 (6–28)	-	7 (6–28)	10 (6–28)	<0.001 *
ALBI score, median (IQR)	−2.80 (−4.39–1.03)	-	−2.98 (−3.76–0.71)	−2.39 (−4.39–1.03)	<0.001 *
Ascites, n (%)	101 (22.9)	-	44 (16.0)	57 (35.2)	<0.001 *
Esophageal varices, n (%)	143 (32.4)	-	79 (30.6)	64 (44.4)	0.007 *
Variceal bleeding, n (%)	27 (6.1)	-	11 (4.0)	16 (9.9)	0.022 *
Albumin, median (IQR), gr/dL	4.2 (1.8–6.4)	-	3.9 (1.8–5.1)	3.7 (2.1–6.4)	<0.001 *
Total bilirubin, median (IQR), mg/dL	0.865 (0.1–29.0)	-	1.1 (0.3–18.2)	1.2 (0.2–29.0)	<0.001 *
Creatinine, median (IQR), mg/dL	0.8 (0.3–4.4)	-	0.7 (0.3–2.0)	0.8 (0.4–4.4)	0.610
Sodium, median (IQR), mEq/L	140 (121–146)	-	140.0 (122.0–146.0)	138.0 (121.0–145.0)	<0.001 *
Platelet count, median (IQR), ×1000/m^3^	164 (28–838)	-	103 (28–430)	147.5 (36–838)	0.041 *
INR, median (IQR)	1.2 (0.9–3.5)	-	1.2 (0.9–3.5)	1.2 (0.9–3.5)	<0.001 *
AFP, median (IQR), ng/mL	3.2 (0.3–200,000.0)	2.1 (0.3–5.8)	2.9 (0.7–253.4)	16.8 (1.1–200,000.0)	<0.001 *
AFP-L3, median (IQR), %	0.5 (0.5–94.4)	0.5 (0.5–0.5)	0.5 (0.5–90.6)	15.9 (0.5–94.4)	<0.001 *
DCP, median (IQR), ng/mL	0.3 (0.1–14,980.0)	0.3 (0.1–0.6)	0.2 (0.1–413.2)	4.8 (0.1–14,980.0)	<0.001 *
GALAD	−3.1 (−9.8–16.7)	−5.0 (−9.8–−1.1)	−3.7 (−8.9–6.1)	2.5 (−4.8–16.7)	<0.001 *
GAAP	−6.1 (−12.6–10.4)	−7.6 (−10.1–−4.1)	−6.6 (−12.6–2.1)	−1.2 (−7.6–10.4)	<0.001 *
ASAP	−5.9 (−10.4–9.8)	−6.7 (−9.0–−4.7)	−6.4 (−10.4–2.0)	−1.7 (−7.0–9.8)	<0.001 *
aMAP	55.3 (19.8–80.3)	44.0 (23.3–64.5)	55.1 (24.7–80.3)	63.4 (19.8–80.2)	<0.001 *
Doylestown	0.2 (0.0–1.0)	0.0 (0.0–0.5)	0.1 (0.0–1.0)	0.8 (0.1–1.0)	<0.001 *

ALBI: Albumin–bilirubin, ALD: Alcohol-associated liver disease; AFP: alfa-fetoprotein; AFP-L3: lens culinaris agglutinin-reactive Alfa-fetoprotein; CTP: Child–Turcotte–Pugh; DCP: des-gamma-carboxy prothrombin; HBV: hepatitis B Virus; HCV: hepatitis C virus; INR: international normalized ratio; IQR: interquartile range; MASLD: metabolic-associated steatotic liver disease; MELD: model for end-stage liver disease; PBC: primary biliary cholangitis. * indicates statistically significant *p*-values as <0.05 among three groups.

**Table 2 cancers-17-02390-t002:** Multivariate logistic regression analysis of serologic biomarkers and composite models for hepatocellular carcinoma occurrence.

Univariate		Multivariate			
	*p*-Value	Beta-Coefficient	aOR	95% CI	*p*-Value
AFP	<0.001 *	0.039	1.040 ^+^	1.016–1.064	<0.001 *
AFP-L3	<0.001 *	0.069	1.071 ^#^	1.051–1.092	<0.001 *
DCP	<0.001 *	0.015	1.015 ˆ	1.003–1.027	0.011 *
GALAD	<0.001 *	0.773	2.166 ^¶^	1.769–2.651	<0.001 *
ASAP	<0.001 *	0.728	2.072 ^†^	1.704–2.519	<0.001 *
GAAP	<0.001 *	0.769	2.157 ^†^	1.759–2.646	<0.001 *
aMAP	<0.001 *	−0.023	0.977 ^¶^	0.935–1.021	0.298
Doylestown	<0.001 *	4.874	130.81 ^⊠^ **^^**	37.257–459.277	<0.001 *

AFP: alpha-fetoprotein; AFP-L3: lens culnaris agglutinin-reactive alpha-fetoprotein; aOR = adjusted odds ratio; DCP: des-gamma-carboxy prothrombin. ^+^ Adjusted for age, gender, diabetes mellitus, hypertension, CTP score, presence of cirrhosis, presence of varices, history of variceal bleeding, AFP-L3, and DCP. ^#^ Adjusted for age, gender, diabetes mellitus, hypertension, CTP score, presence of cirrhosis, presence of varices, history of variceal bleeding, AFP, and DCP. ˆ Adjusted for age, gender, diabetes mellitus, hypertension, CTP score, presence of cirrhosis, presence of varices, history of variceal bleeding, AFP, and AFP-L3. ^¶^ Adjusted for diabetes mellitus, hypertension, CTP score, presence of cirrhosis, presence of varices, history of variceal bleeding. ^†^ Adjusted for diabetes mellitus, hypertension, CTP score, presence of cirrhosis, presence of varices, history of variceal bleeding, and AFP-L3. **^^**
^⊠ ^Adjusted for diabetes mellitus, hypertension, CTP score, presence of cirrhosis, presence of varices, history of variceal bleeding, AFP-L3, and DCP. * indicates *p* value < 0.05.

**Table 3 cancers-17-02390-t003:** Performance of serological biomarkers and composite models in the detection of HCC with optimal cut-off values determined by the Youden index.

	Any-Stage HCC (*n* = 165)	Early-Stage HCC (*n* = 64)	AFP-Negative HCC (*n* = 88)	Viral HCC (*n* = 110)	Non-Viral HCC (*n* = 55)
Optimal cut-off					
AFP	6.15	5.25	-	6.30	13.20
AFP-L3	0.90	0.95	7.15	0.95	9.95
DCP	0.35	0.35	0.45	0.55	1.15
GALAD	−1.96	−2.55	−1.70	−1.70	−0.43
GAAP	−5.09	−5.49	−5.09	−4.9	−1.81
ASAP	−5.32	−5.67	−5.17	−5.0	−4.19
aMAP	60.19	60.44	61.96	60.20	67.39
Doylestown	0.33	0.23	0.38	0.48	0.63
Sensitivity, (%)					
AFP	72.1 (64.6–78.8)	64.1 (51.1–75.7)	-	73.6 (64.4–81.6)	63.6 (49.6–76.2)
AFP-L3	82.4 (75.7–87.9)	73.4 (60.9–83.7)	58.0 (47.0–68.4)	82.7 (74.4–89.3)	61.8 (47.7–74.6)
DCP	85.5 (79.13–90.45)	73.4 (60.9–83.7)	71.6 (61.0–80.7)	80.0 (71.3–87.0)	70.9 (57.1–82.4)
GALAD	90.3 (84.73–94.36)	89.1 (78.6–95.5)	80.7 (70.9–88.3)	92.7 (86.2–96.8)	72.7 (59.0–83.9)
GAAP	88.8 (82.8–93.2)	85.7 (74.6–93.3)	84.7 (75.3–91.6)	87.0 (79.2–92.7)	59.6 (45.1–73.0)
ASAP	87.5 (81.4–92.2)	84.1 (72.7–92.1)	77.4 (67.3–86.0)	87.0 (79.2–92.7)	71.2 (56.9–82.9)
aMAP	72.8 (65.3–79.5)	79.4 (67.3–88.5)	57.0 (45.9–67.6)	70.6 (61.2–79.0)	37.7 (24.8–52.1)
Doylestown	83.8 (77.1–89.1)	76.2 (63.8–86.0)	72.9 (62.2–82.0)	70.4 (60.8–78.8)	71.2 (56.9–82.9)
Specificity, (%)					
AFP	84.8 (80.1–88.8)	78.7 (73.4–83.4)	-	86.1 (80.5–90.5)	94.7 (87.1–98.6)
AFP-L3	80.5 (75.3–85.0)	80.5 (75.3–85.0)	84.6 (79.7–88.6)	85.1 (79.4–89.7)	86.8 (77.1–93.5)
DCP	78.7 (73.40–83.37)	78.7 (73.4–83.4)	87.5 (83.0–91.2)	94.5 (90.4–97.2)	84.2 (74.0–91.6)
GALAD	80.5 (75.3–85.0)	73.6 (68.0–78.7)	86.0 (81.3–89.9)	88.6 (83.3–92.6)	88.2 (78.7–94.4)
GAAP	83.1 (80.0–87.5)	76.5 (70.8–81.5)	84.1 (78.9–88.4)	91.4 (86.5–95.0)	95.6 (87.6–99.1)
ASAP	82.0 (76.7–86.5)	76.1 (70.3–81.2)	85.3 (80.3–89.4)	94.1 (89.7–97.0)	83.8 (72.9–91.6)
aMAP	67.0 (61.1–72.6)	67.4 (61.5–73.0)	72.9 (67.2–78.2)	79.1 (72.7–84.6)	73.0 (61.4–82.7)
Doylestown	80.0 (74.6–84.7)	71.8 (65.8–77.2)	83.7 (78.5–88.0)	95.2 (91.1–97.8)	83.8 (72.9–91.6)
PPV, (%)					
AFP	73.9 (67.8–79.1)	41.0 (34.2–48.2)	-	74.3 (66.8–80.6)	89.7 (76.6–95.9)
AFP-L3	71.5 (66.2–76.4)	46.5 (39.7–53.6)	54.8 (46.6–62.8)	75.2 (68.3–81.0)	77.3 (64.8–86.3)
DCP	70.5 (65.4–75.1)	44.3 (37.8–51.1)	65.0 (56.9–72.3)	88.9 (81.7–93.5)	76.5 (65.3–84.9)
GALAD	73.4 (68.4–77.9)	43.9 (38.7–49.2)	65.1 (57.8–71.9)	81.6 (75.1–86.7)	81.6 (70.2–89.3)
GAAP	76.7 (71.4–81.4)	47.4 (41.4–53.4)	64.3 (57.2–70.8)	85.5 (78.5–90.4)	91.2 (77.0–97.0)
ASAP	75.3 (70.0–79.9)	46.5 (40.5–52.6)	64.1 (56.5–71.0)	89.5 (82.7–93.8)	77.1 (65.6–85.6)
aMAP	57.0 (52.2–61.7)	36.2 (31.5–41.3)	41.0 (34.2–47.1)	65.3 (58.2–71.7)	50.0 (37.5–62.5)
Doylestown	72.4 (67.1–77.2)	40.0 (34.4–45.9)	60.2 (52.6–67.3)	89.4 (81.5–94.2)	77.1 (65.6–85.6)
NPV, (%)					
AFP	83.6 (79.9–86.7)	90.5 (87.2–92.9)	-	85.6 (81.3–89.1)	78.3 (71.7–83.7)
AFP-L3	88.5 (84.6–91.5)	92.9 (89.7–95.2)	86.1 (82.9–88.9)	90.0 (85.6–93.2)	75.9 (69.0–81.7)
DCP	90.1 (86.2–92.9)	92.8 (89.5–95.1)	90.5 (87.2–93.0)	89.6 (85.6–92.6)	80.0 (72.4–85.9)
GALAD	93.3 (89.7–95.7)	96.7 (93.5–98.3)	93.2 (90.0–95.5)	95.7 (91.9–97.8)	81.7 (74.2–87.4)
GAAP	92.2 (88.4–94.8)	95.6 (92.2–97.6)	94.2 (90.8–96.4)	92.4 (88.2–95.2)	75.6 (68.9–81.2)
ASAP	91.3 (87.3–94.1)	95.1 (91.6–97.2)	91.9 (88.3–94.4)	92.6 (88.5–95.4)	79.2 (71.0–85.5)
aMAP	80.4 (75.9–84.3)	93.3 (89.5–95.8)	84.0 (80.3–87.1)	82.9 (78.2–86.7)	62.1 (56.0–67.8)
Doylestown	88.7 (84.6–91.9)	92.4 (88.6–95.0)	90.1 (86.5–92.9)	84.8 (80.6–88.2)	79.2 (71.0–85.5)

AFP: alpha-fetoprotein; AFP-L3: lens culinaris agglutinin-reactive alpha-fetoprotein; DCP: des-gamma-carboxy prothrombin, HCC: hepatocellular carcinoma, NPV: negative predictive value, PPV: positive predictive value.

**Table 4 cancers-17-02390-t004:** Performance of serological biomarkers and composite models in the detection of HCC with established cut-offs.

	Any-Stage HCC (n = 165)	Early-Stage HCC (n = 64)	AFP-Negative HCC (n = 88)	Viral HCC (n = 110)	Non-Viral HCC (n = 55)
Established cut-off					
AFP	20	20	-	20	20
AFP-L3	10	10	10	10	10
DCP	7.5	7.5	7.5	7.5	7.5
GALAD	−0.63	−0.63	−0.63	−0.63	−0.63
GAAP	−0.65	−0.65	−0.65	−0.65	−0.65
ASAP	0.56	0.56	0.56	0.56	0.56
aMAP	60	60	60	60	60
Doylestown	0.5	0.5	0.5	0.5	0.5
Sensitivity, (%)					
AFP	46.7 (38.9–54.6)	34.4 (23.0–47.3)	-	47.3 (37.7–57.0)	45.5 (32.0–59.5)
AFP-L3	61.2 (53.3–68.7)	46.9 (34.3–59.8)	44.3 (33.7–55.3)	60.9 (51.1–70.1)	61.8 (47.7–74.6)
DCP	43.6 (35.9–51.6)	15.6 (7.8–26.9)	31.8 (22.3–42.6)	44.6 (35.1–54.3)	41.8 (28.7–55.9)
GALAD	75.8 (68.5–82.1)	57.8 (44.8–70.1)	58.0 (47.0–68.4)	76.4 (67.3–83.9)	74.6 (61.0–85.3)
GAAP	43.8 (35.9–51.8)	19.1 (10.3–30.9)	24.7 (16.0–35.3)	41.7 (32.3–51.6)	48.1 (31.0–62.4)
ASAP	32.5 (25.3–40.3)	12.7 (5.7–23.5)	16.5 (9.3–26.1)	32.4 (23.7–42.1)	32.7 (20.3–47.1)
aMAP	72.8 (65.3–79.5)	79.4 (67.3–88.5)	72.1 (61.4–81.2)	70.6 (61.2–79.0)	77.4 (63.8–87.7)
Doylestown	71.3 (63.6–78.1)	54.0 (40.9–66.6)	58.8 (47.6–69.4)	69.4 (59.8–78.0)	75.0 (61.1–86.0)
Specificity, (%)					
AFP	98.2 (95.8–99.4)	98.2 (95.8–99.4)	-	99.5 (97.3–100.0)	94.7 (87.1–98.6)
AFP-L3	89.2 (84.9–92.6)	89.2 (84.9–92.6)	90.4 (86.3–93.7)	90.1 (85.1–93.8)	86.8 (77.1–93.5)
DCP	97.5 (94.9–99.0)	97.5 (94.9–99.0)	98.2 (95.8–99.4)	98.5 (95.7–99.7)	94.7 (87.1–98.6)
GALAD	93.5 (89.9–96.1)	93.5 (89.9–96.1)	94.9 (91.5–97.2)	96.5 (93.0–98.6)	85.5 (75.6–92.6)
GAAP	98.8 (96.6–99.8)	98.8 (96.6–99.8)	99.6 (97.8–100.0)	99.5 (97.1–100.0)	97.1 (89.8–99.6)
ASAP	99.2 (97.2–99.9)	99.2 (97.2–99.9)	99.6 (97.8–100.0)	99.5 (97.1–100.0)	98.5 (92.1–100.0)
aMAP	65.9 (59.9–71.6)	65.9 (59.9–71.6)	66.5 (60.5–72.2)	77.6 (71.1–83.2)	35.1 (24.9–47.1)
Doylestown	89.0 (84.5–92.6)	89.0 (84.5–92.6)	89.6 (85.2–93.1)	95.7 (91.7–98.1)	70.6 (58.3–81.0)
PPV, (%)					
AFP	93.9 (86.4–97.4)	81.5 (63.4–91.8)	-	98.1 (87.9–99.7)	86.2 (69.8–94.4)
AFP-L3	77.1 (70.2–82.8)	50.0 (39.5–60.5)	60.0 (49.3–69.8)	77.0 (68.3–83.9)	77.3 (64.8–86.3)
DCP	91.1 (82.9–95.6)	58.8 (36.1–78.3)	84.9 (69.0–93.4)	94.2 (83.9–98.1)	85.2 (67.8–94.0)
GALAD	87.4 (81.5–91.6)	67.3 (55.7–77.1)	78.5 (68.0–86.2)	92.3 (85.2–96.2)	78.9 (67.9–86.8)
GAAP	95.9 (88.2–98.7)	80.0 (53.8–93.2)	95.5 (74.2–99.4)	97.8 (86.3–99.7)	92.6 (75.6–98.1)
ASAP	96.3 (86.5–99.1)	80.0 (46.5–94.8)	93.3 (65.1–99.1)	97.2 (82.9–99.6)	94.4 (70.0–99.2)
aMAP	56.2 (51.5–60.8)	35.2 (30.6–40.1)	41.1 (36.0–46.3)	63.6 (56.8–70.0)	46.1 (40.6–51.6)
Doylestown	80.3 (73.9–85.4)	54.8 (44.4–64.8)	65.8 (56.2–74.3)	90.4 (82.5–94.9)	66.1 (56.7–74.4)
NPV, (%)					
AFP	75.6 (72.8–78.1)	86.6 (84.4–88.6)	-	77.5 (74.3–80.5)	70.6 (65.2–75.4)
AFP-L3	79.4 (76.0–82.4)	87.9 (85.2–90.2)	83.4 (80.6–85.9)	88.8 (76.8–84.2)	75.9 (69.0–81.7)
DCP	74.4 (71.7–76.9)	83.3 (81.8–84.8)	81.7 (79.4–83.7)	76.5 (73.3–79.3)	69.2 (64.1–73.9)
GALAD	86.6 (83.2–89.5)	90.6 (87.8–92.8)	87.5 (84.5–89.9)	88.2 (84.2–91.3)	82.3 (74.5–88.1)
GAAP	73.7 (70.9–76.3)	83.2 (81.4–84.8)	79.6 (77.6–81.5)	74.7 (71.6–77.6)	71.0 (65.2–76.1)
ASAP	70.1 (67.8–72.3)	82.1 (80.7–83.5)	77.9 (76.2–79.5)	71.8 (69.1–74.4)	65.7 (61.3–69.9)
aMAP	80.2 (75.6–84.1)	93.2 (89.3–95.7)	88.1 (83.9–91.3)	82.6 (77.9–86.5)	68.4 (54.7–79.6)
Doylestown	83.2 (79.4–86.3)	88.7 (85.7–91.1)	86.5 (83.3–89.3)	84.4 (80.3–87.8)	78.7 (69.2–85.8)

AFP: alpha-fetoprotein; AFP-L3: lens culinaris agglutinin-reactive alpha-fetoprotein; DCP: des-gamma-carboxy prothrombin, HCC: hepatocellular carcinoma, NPV: negative predictive value, PPV: positive predictive value.

**Table 5 cancers-17-02390-t005:** Performance of serological biomarkers and composite models in the detection of HCC with cut-offs providing 90% specificity.

	Any-Stage HCC (n = 165)	Early-Stage HCC (n = 64)	AFP-Negative HCC (n = 88)	Viral HCC (n = 110)	Non-Viral HCC (n = 55)
90% specificity cut-off					
AFP	8.70	8.80	-	8.25	9.80
AFP-L3	11.65	11.65	10.50	11.15	16.45
DCP	0.65	0.65	0.55	0.45	3.15
GALAD	−1.22	−1.22	−1.31	−1.35	−0.02
GAAP	−4.27	−4.24	−4.38	−5.06	−3.02
ASAP	−4.64	−4.63	−4.81	−5.52	−2.86
aMAP	69.11	69.11	68.67	66.58	71.17
Doylestown	0.52	0.52	0.51	0.38	0.73
Sensitivity, (%)					
AFP	63.0 (55.2–70.4)	48.4 (35.8–61.3)	-	62.7 (53.0–71.8)	63.6 (49.6–76.2)
AFP-L3	57.6 (49.7–65.2)	40.6 (28.5–53.6)	43.2 (32.7–54.2)	59.1 (49.3–68.4)	49.1 (35.4–62.9)
DCP	74.6 (67.2–81.0)	59.4 (46.4–71.5)	67.1 (65.2–76.7)	81.8 (73.3–88.5)	52.7 (38.8–66.4)
GALAD	81.2 (74.4–86.9)	68.8 (55.9–79.8)	68.2 (57.4–77.7)	83.6 (75.4–90.0)	69.1 (55.2–80.9)
GAAP	77.5 (70.2–83.7)	60.3 (47.2–72.4)	65.9 (54.8–75.8)	88.9 (81.4–94.1)	65.4 (50.9–78.0)
ASAP	80.0 (73.0–85.9)	66.7 (53.7–78.1)	70.6 (59.7–80.0)	92.6 (85.9–96.8)	59.6 (45.1–73.0)
aMAP	22.2 (16.1–29.4)	22.2 (12.7–34.5)	22.1 (13.9–32.3)	28.4 (20.2–38.9)	28.3 (16.8–42.4)
Doylestown	70.0 (62.3–77.0)	52.4 (39.4–65.1)	57.8 (46.5–68.3)	80.6 (71.8–87.5)	63.5 (49.0–76.4)
Specificity, (%)					
AFP	90% (89.5–90.5%)	90% (89.5–90.5%)	-	90% (89.5–90.5%)	90% (89.5–90.5%)
AFP-L3	90% (89.5–90.5%)	90% (89.5–90.5%)	90% (89.5–90.5%)	90% (89.5–90.5%)	90% (89.5–90.5%)
DCP	90% (89.5–90.5%)	90% (89.5–90.5%)	90% (89.5–90.5%)	90% (89.5–90.5%)	90% (89.5–90.5%)
GALAD	90% (89.5–90.5%)	90% (89.5–90.5%)	90% (89.5–90.5%)	90% (89.5–90.5%)	90% (89.5–90.5%)
GAAP	90% (89.5–90.5%)	90% (89.5–90.5%)	90% (89.5–90.5%)	90% (89.5–90.5%)	90% (89.5–90.5%)
ASAP	90% (89.5–90.5%)	90% (89.5–90.5%)	90% (89.5–90.5%)	90% (89.5–90.5%)	90% (89.5–90.5%)
aMAP	90% (89.5–90.5%)	90% (89.5–90.5%)	90% (89.5–90.5%)	90% (89.5–90.5%)	90% (89.5–90.5%)
Doylestown	90% (89.5–90.5%)	90% (89.5–90.5%)	90% (89.5–90.5%)	90% (89.5–90.5%)	90% (89.5–90.5%)
PPV, (%)					
AFP	79.4 (72.5–84.9)	53.5 (42.6–64.0)	-	78.4 (69.8–85.1)	81.4 (68.8–89.7)
AFP-L3	78.5 (71.2–84.4)	50.0 (38.4–61.6)	60.3 (49.4–70.3)	78.3 (69.4–85.2)	77.1 (62.4–87.3)
DCP	83.1 (77.0–87.8)	60.3 (49.8–69.9)	68.6 (59.8–76.3)	85.7 (78.5–90.8)	78.4 (64.3–88.0)
GALAD	83.2 (77.5–87.7)	62.0 (52.3–70.7)	69.0 (60.2–76.6)	83.6 (76.6–88.9)	82.6 (70.7–90.4)
GAAP	83.2 (77.2–87.9)	60.3 (49.9–69.9)	69.1 (60.0–77.0)	84.2 (77.4–89.3)	82.9 (70.1–91.0)
ASAP	83.7 (77.8–88.2)	62.7 (52.7–71.7)	70.6 (61.8–78.1)	84.8 (78.1–89.6)	81.6 (68.0–90.2)
aMAP	58.1 (46.5–68.8)	35.0 (23.0–49.3)	42.2 (29.9–55.6)	62.0 (49.2–73.3)	65.2 (46.2–80.4)
Doylestown	80.6 (74.1–85.7)	55.0 (44.4–65.2)	65.3 (55.6–79.9)	81.3 (74.0–86.9)	82.5 (69.4–90.7)
NPV, (%)					
AFP	80.4 (77.0–83.4)	88.3 (85.6–90.6)	-	81.6 (77.6–85.0)	77.3 (70.4–83.0)
AFP-L3	78.2 (74.9–81.1)	86.9 (84.3–89.0)	83.2 (80.4–85.6)	80.3 (76.4–83.6)	70.8 (64.9–76.1)
DCP	85.7 (82.2–88.7)	90.7 (87.8–92.9)	89.4 (86.2–91.9)	90.3 (86.2–93.3)	72.3 (66.2–77.8)
GALAD	89.0 (85.4–91.7)	92.6 (89.7–94.7)	89.7 (86.5–92.3)	91.0 (86.9–94.0)	80.0 (72.8–85.7)
GAAP	86.5 (82.7–89.5)	90.2 (87.1–92.6)	88.6 (85.3–91.3)	93.4 (89.2–96.0)	77.3 (69.8–83.2)
ASAP	87.8 (84.0–90.8)	91.6 (88.5–94.0)	90.0 (86.6–92.7)	95.5 (91.5–97.6)	74.4 (67.4–80.3)
aMAP	66.0 (83.9–68.0)	83.3 (81.3–85.1)	78.2 (76.1–80.1)	69.4 (66.7–72.0)	63.5 (59.0–67.7)
Doylestown	82.6 (79.0–85.8)	88.4 (85.4–90.8)	86.2 (82.9–88.9)	88.8 (84.4–92.1)	76.3 (69.0–82.3)

AFP: alpha-fetoprotein; AFP-L3: lens culinaris agglutinin-reactive alpha-fetoprotein; DCP: des-gamma-carboxy prothrombin, HCC: hepatocellular carcinoma, NPV: negative predictive value, PPV: positive predictive value.

## Data Availability

The data to support the results reported in this study are available from the corresponding author upon reasonable request.

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
