# Peer review of "Enhancing Hepatocellular Carcinoma Surveillance: Comparative Evaluation of AFP, AFP-L3, DCP and Composite Models in a Biobank-Based Case-Control Study"

_cancers, 2025, doi:10.3390/cancers17142390_

Round 1
Reviewer 1 Report
Comments and Suggestions for Authors
Comments to the Authors:
In this work, Coskun Demirtas and colleagues used robust statistical analysis to compare the diagnostic performance of some established serological biomarkers versus composite models for HCC detection in a Biobank-Based Case-Control Study with an aim to enhance HCC Surveillance.
Below, I have outlined some points for the authors to address:
- Including normal (control) subjects, which are neither HCC patients nor patients with chronic liver disease will enhance the robustness of the study.
- Although the study compared only three (3) serum biomarkers to the composite models. Evaluation of the diagnostic ability other biomarkers (especially Glypican-3 and Osteopontin) in comparison to composite models is necessary.
- In the exclusion criteria, it was not clearly stated if patients on medication (that may affect the levels of the biomarkers), especially for the CLD groups were excluded from the study.
- The authors should be consistent with the use of “alpha-fetoprotein” or “alfa-fetoprotein” throughout the manuscript. Either of them is correct.
- Graphs in Figures 1& 2 including axes labeling, should be enlarged for clarity and readability.
Author Response
We are pleased to resubmit our manuscript (cancers-3737486) entitled ‘’Enhancing Hepatocellular Carcinoma Surveillance: Comparative Evaluation of AFP, AFP-L3, DCP and Composite Models in a Biobank-Based Case-Control Study’’. We found your critique of our initial submission to be very helpful and changed the manuscript in consistent with your suggestions. Our point-by-point responses to the critique are outlined below.
Reviewer 1
Comments to the Authors:
In this work, Coskun Demirtas and colleagues used robust statistical analysis to compare the diagnostic performance of some established serological biomarkers versus composite models for HCC detection in a Biobank-Based Case-Control Study with an aim to enhance HCC Surveillance.
Below, I have outlined some points for the authors to address:
Comments 1: Including normal (control) subjects, which are neither HCC patients nor patients with chronic liver disease will enhance the robustness of the study.
Response 1: Thank you for your valuable comment and suggestion. In response, we have included a healthy control group consisting of individuals with no evidence or history of any health condition, including chronic liver disease (CLD) or hepatocellular carcinoma (HCC), as confirmed by clinical records and relevant laboratory data. The inclusion of this group allowed us to define baseline biomarker levels in a healthy population and provided additional context for interpreting biomarker elevation. However, we used the CLD group as the reference control group when evaluating risk and diagnostic performance, as these biomarkers and models are specifically intended for use in CLD patients who are at high risk of developing HCC. We have revised the Simple Summary (Page 1, Line 23-24), Abstract (Page 1, Line 40-41), Materials and Methods (Page 4, Line 124-125 and 129-131) and Results (Page 6, Line 221-225, Line 246-253, and Line 274-281) sections accordingly and have integrated the new analyses into the revised manuscript.
The following statement was added to the Materials and Methods section (Page 4, Line 124-125 and 129-131):
- The study cohort comprised 562 adult participants, including 120 healthy control participants, 277 patients with CLD at risk of HCC, and 165 patients with HCC.
- Although the main focus of this study was the comparison between HCC and CLD groups, a healthy control group was included to establish baseline levels of biomarkers and composite models in a healthy population.
Comments 2: Although the study compared only three (3) serum biomarkers to the composite models. Evaluation of the diagnostic ability other biomarkers (especially Glypican-3 and Osteopontin) in comparison to composite models is necessary.
Response 2: Thank you for your valuable comment. We agree that including a broader range of biomarkers, such as Glypican-3 and Osteopontin, could provide additional perspective. However, the primary aim of our study was not to comprehensively evaluate all promising serological biomarkers for HCC, but rather to compare the diagnostic performance of widely recognized and clinically proposed composite models —GALAD, GAAP, ASAP, aMAP, and Doylestown— against traditional AFP-based screening.
To contextualize these models, we also assessed the individual diagnostic performance of the key biomarkers incorporate within these models —AFP, AFP-L3, and DCP. Therefore, our comparison was focused specifically on biomarkers that are integral components of these composite scores, rather than exploring on the full spectrum of emerging biomarkers of HCC.
Nonetheless, we acknowledge that this may have caused some confusion regarding the scope of the study. To address this, we have revised the manuscript title to ‘’Enhancing Hepatocellular Carcinoma Surveillance: Comparative Evaluation of AFP, AFP-L3, DCP, and Composite Models in a Biobank-Based Case-Control Study’’. Additionally, we have clarified this point in the Discussion section under the limitation paragraph with the following statement (Page 17, Line 504-510);
- Thirdly, our study did not include other emerging biomarkers such as Glypican-3 and osteopontin, which is a limitation worth noting. These biomarkers may offer additional diagnostic value; however, our focus was to compare established composite models and the core biomarkers they rely on, rather than to perform a broad evaluation of all potential serologic biomarkers for HCC.
Comments 3: In the exclusion criteria, it was not clearly stated if patients on medication (that may affect the levels of the biomarkers), especially for the CLD groups were excluded from the study.
Response 3: Thank you for your valuable comment. All HCC patients included in our study were newly diagnosed and had not received any HCC-specific treatments or medications at the time of specimen collection. Furthermore, none of the participants included in the study were receiving medications known to affect the levels of AFP, AFP-L3, or DCP, such as vitamin K antagonists, HCC-specific therapies including chemotherapy/immunotherapy, interferon, or any other immunosupressants. To clarify this point, we have added the following sentence to the Materials and Methods section (Page 4, Line 138-141) as follows;
- No subjects included in the study were receiving medications known to affect AFP, AFP-L3, or DCP levels —such as vitamin K antagonists or active HCC-specific therapies— to avoid confounding effects on biomarker measurements.
Comments 4: The authors should be consistent with the use of “alpha-fetoprotein” or “alfa-fetoprotein” throughout the manuscript. Either of them is correct.
Response 4: Thank you for your helpful comment. We have carefully reviewed the manuscript and ensured consistent use of the term ‘’alpha-fetoprotein’’ throughout the manuscript, in accordance with your suggestion.
Comments 5: Graphs in Figures 1& 2 including axes labeling, should be enlarged for clarity and readability.
Response 5: Thank you for your valuable suggestion. To improve clarity and readability, we have enlarged the graphs including the axes labeling. Additionally, we simplified Figure 1 to make it more concise and divided the original Figure 2 into two separate figures, now presented as Figure 2 and Figure 3, in the revised manuscript.

Reviewer 2 Report
Comments and Suggestions for Authors
From a biostats and clinical epidemiology point of view, here are some comments for the authors:
From a biostats and clinical epidemiology point of view, here are some comments for the authors:
- ref #2, update it to Globocan 2022 and add the GBD 2021 study estimates for HCC, both worldwide and in Turkey (afterwards, show any potential difference between these 2 geographical entities!)
- stats analyses, always use an inferential non-parametric approach, currently no inferential tests have been performed
- stats analyses, a full series of uni/multivariate binary logistic models is needed, having as outcome the HCC occurrence; this is the most critical step, since currently huge selection biases do exist between controls and cases groups, and they must be controlled
- figure 1, hard to be interpreted, simplify it
- table 1, add an overall column, on the left side of the table
- tables 3-4-5, poorly informative, you need a dedicated non linear model, as above stated
- conclusion, the lack of predictive models prevent any stable conclusion
Author Response
We are pleased to resubmit our manuscript (cancers-3737486) entitled ‘’Enhancing Hepatocellular Carcinoma Surveillance: Comparative Evaluation of AFP, AFP-L3, DCP and Composite Models in a Biobank-Based Case-Control Study’’. We found your critique of our initial submission to be very helpful and changed the manuscript in consistent with your suggestions. Our point-by-point responses to the critique are outlined below.
Comments and Suggestions for Authors
From a biostats and clinical epidemiology point of view, here are some comments for the authors:
Comments 1: ref #2, update it to Globocan 2022 and add the GBD 2021 study estimates for HCC, both worldwide and in Turkey (afterwards, show any potential difference between these 2 geographical entities!)
Response 1: Thank you for your suggestion and contribution. We revised the introduction part to incorporate the most recent data from both the GLOBOCAN 2022 and GBD 2021 studies, as recommended . The revised text is included in the Introduction section (Page 2, Line 63-70) as follows;
- Hepatocellular carcinoma (HCC), the predominant form of primary liver cancer, is the sixth most common cancer worldwide and the third leading cause of cancer-related deaths[1]. According to Global Burden of Disease (GBD) 2021 study, there were approximately 529,202 incident HCC cases and 483,875 deaths related to liver cancer, representing increases of 25% and 26%, respectively, between 2010 to 2021[2]. In Turkey, an estimated 5,039 new HCC cases and 4,929 related deaths were reported in 2022, reflecting a notable national burden in line with global trends[3]. These findings underscore the importance of continued efforts in HCC prevention, early detection, and effective management strategies.
Comments 2: stats analyses, always use an inferential non-parametric approach, currently no inferential tests have been performed
Response 2: Thank you for your valuable observation. We actually used appropriate inferential non-parametric tests throughout our analyses, but it was not stated correctly in the Statistical Analyses section of the Methods, in the manuscript. We have now revised this section to explicitly describe the use of non-parametric inferential tests to ensure clarity (Page 5, Line 173-179) as follows;
- All group comparisons and diagnostic performance analyses were conducted using appropriate non-parametric inferential tests to ensure robust and valid statistical inference. Differences between groups were assessed using the chi-square test of Fisher’s exact test for categorical variables, as appropriate based on expected cell counts; and the Mann-Whitney U test for comparisons between two groups, and the Kruskal-Wallis test for comparisons among three groups for continuous variables.
Comments 3: stats analyses, a full series of uni/multivariate binary logistic models is needed, having as outcome the HCC occurrence; this is the most critical step, since currently huge selection biases do exist between controls and cases groups, and they must be controlled
Response 3: Thank you for this important and constructive comment. In response, we performed a full series of univariate binary logistic regression analyses using HCC occurrence as the outcome variable. These analyses were conducted to identify significant predictors of HCC and to address potential selection biases between the HCC and CLD groups. To avoid redundancy and collinearity, individual variables such as albumin, bilirubin, and INR were not included in the regression models, as they are components of the Child-Pugh score, which was included as a covariate in the analyses. The full results of the univariate analyses are provided in Supplementary Table 3, and statistically significant findings have been incorporated into the Results section (Page 9, Line 295-298) of the revised manuscript.
Based on the univariate analysis’s findings, we then performed multiple multivariable logistic regression models, each including only those variables that were statistically significant in univariate analysis. We did not include all serological biomarkers and composite models together in a single multivariable model, since the composite models (e.g., GALAD, GAAP, ASAP) already incorporate variables such as AFP, AFP-L3, DCP, age and gender, that were statistically significant in univariate analyses. Including both the models and their component variables in the same analysis would introduce multicollinearity and reduce model validity.
To clearly present these results, we have added a new Table 2 to the revised manuscript, which reports the adjusted odds ratios (aORs), 95% confidence intervals, beta-coefficients, and p-values for each multivariable model (Page 9-10, Line 309-333). The specific covariates included in each model are detailed in the table footnotes using corresponding symbols for clarification and transparency. We summarized the binary logistic regression analyses findings in the Results section (Page 9, Line 291-305) in the revised manuscript.
We believe these additions significantly strengthen the analytical robustness of the study and adequately address the concerns regarding potential selection bias. We added the following part to the Statistical Analyses part of Materials and Methods section (Page 5, Line 180-188):
- To evaluate predictors of HCC occurrence, univariate binary logistic regression analyses were initially conducted for all variables including each biomarker and composite model. Variables demonstrating statistical significance in univariate analyses were subsequently included in multiple multivariable logistic regression models to identify independent association with HCC. To prevent multicollinearity, each multivariable model incorporated only one biomarker or composite model, alongside covariates that did not overlap with components of the included biomarker or model. Adjusted odds ratios with 95% confidence intervals, beta-coefficients, and p-values were reported for all regression models.
Comments 4: figure 1, hard to be interpreted, simplify it
Response 4: Thank you for your helpful suggestion. We have simplified Figure 1, by removing the arrows previously placed above the bars, which were potentially cluttering the visualization. Instead, we now used asterisks to indicate a statistically significant differences (p<0.05) in comparisons between any-stage or early stage, and the reference group (CLD). Additionally, to further reduce complexity and enhance focus on clinically relevant stages for surveillance, we removed the advanced-stage HCC bar from the figure.
We believe these modifications significantly improve the figure’s readability and overall clarity.
Comments 5: table 1, add an overall column, on the left side of the table
Response 5: Thank you for your valuable suggestion. We have revised Table 1 by adding an ‘’Overall Cohort’’ column on the left side to present baseline characteristics, laboratory, biomarker levels and composite model scores across the entire study cohort. In addition, based on another reviewer’s recommendation, we incorporated a new comparison group — healthy controls — as presented their data alongside those of the CLD and HCC groups. The revised Table 1 now includes a comprehensive overview of all three groups: healthy controls, CLD, and HCC, and the Overall column, on the left side of the table. To maintain clarity and prevent overcrowding, detailed pairwise comparisons between specific groups are provided separately in a newly created Supplementary Table 1 in the revised manuscript.
Comments 6: tables 3-4-5, poorly informative, you need a dedicated nonlinear model, as above stated
Response 6: Thank you for your valuable comment. We acknowledge the need for a non-linear model and have addressed this by presenting dedicated non-linear models in Table 2 of the revised manuscript. However, we retained Tables 3, 4 and 5 because they illustrate the superior performance metrics of the GALAD model across different cut-off approaches. Given the ongoing debate in the literature about the optimal GALAD cut-off, these tables demonstrate that GALAD consistently outperforms other models in HCC detection regardless of cut-off selection. To avoid an excessive number of tables in the revised manuscript, we moved the previous Table 2, titled ‘’Characteristics of patients with HCC, to Supplementary Table 1.
Comments 7: conclusion, the lack of predictive models prevent any stable conclusion
Response 7: Thank you for your valuable comment. As demonstrated in our binary logistic regression analysis, all the serological biomarkers and the composite models investigated in our study were independently associated with the HCC risk. Therefore, we conducted the sensitivity, specificity, and AUC analysis, presented in table 3, 4, and 5. Our results showed consistent performance of the GALAD score across diverse populations and highlighted its superiority over both individual biomarkers and other composite models. In the era of modern combined serological biomarker based HCC screening, we conclude that the GALAD score currently offers the highest diagnostic value among available tools.
However, GAAP and ASAP scores — which use one fewer biomarker (AFP-L3) — demonstrated comparable performance, particularly in patients with viral etiology. These findings suggest that GAAP and ASAP could serve as cost-effective alternatives in resource-limited settings or in populations with viral etiology dominance.
In contrast, reliance on single biomarkers, as well as the aMAP and Doylestown scores, is associated with lower diagnostic accuracy and may result in less robust screening performance in clinical practice. Therefore, we recommend the GALAD score as the preferred clinical tool for HCC screening, while acknowledging that GAAP and ASAP may be suitable alternatives in patients with viral etiology.
Recognizing the need for clearer conclusions, we have revised the Conclusion section accordingly (Page 17, Line 512-523) in the revised manuscript;
-Our results demonstrate the consistent performance of the GALAD score across diverse populations and underscore its superiority over individual biomarkers and other composite models. Notably, the GAAP and ASAP scores—which use one fewer biomarker (AFP-L3) —exhibited comparable performance, particularly in viral etiology. These results suggest that GAAP and ASAP may serve as cost-effective alternatives in resource constrained settings or where viral hepatitis dominates. In contrast, reliance on single biomarkers, as well as the use of aMAP and the Doylestown algorithm, is associated with lower diagnostic accuracy and may compromise screening effectiveness in clinical practice. Based on these findings, the GALAD score is recommended as the current preferred screening tool for HCC, while prospective validation of established and optimized cut-off values in larger, multi-ethnic, real-world cohorts is warranted.
We also revised the conclusion part of the abstract as follows (Page 2, Line 52-56);
-Our results demonstrate the consistent performance of the GALAD score across diverse populations and underscore its superiority over individual biomarkers and other composite models. Notably, the GAAP and ASAP scores—which use one fewer biomarker (AFP-L3) —exhibited comparable performance, particularly in viral etiology.

Round 2
Reviewer 2 Report
Comments and Suggestions for Authors
Teo more comments, about binary logistic models:
1- add the univariate serires results too
2- state the number of events included in the multivariate model
Author Response
We are pleased to resubmit our manuscript (cancers-3737486) entitled ‘’Enhancing Hepatocellular Carcinoma Surveillance: Comparative Evaluation of AFP, AFP-L3, DCP and Composite Models in a Biobank-Based Case-Control Study’’. We found your additional comments helpful and have changed the manuscript in consistent with your suggestions. Our point-by-point responses to the critique are outlined below.
Comment 1: Add the univariate series results too.
Response 1: Thank you for your suggestion. We included the results of the univariate binary logistic regression for each parameter that was considered as potential parameters for the multivariate model. The statistically significant parameters identified in the univariate analyses are summarized in the Results section in Page 8-9, Line 290-293, and presented in detail in revised version of Supplementary Table 3, including beta-coefficients, Odds ratios, 95% confidence intervals and p-values.
Comment 2: State the number of events included in the multivariate model.
Response 2: Thank you for your helpful comment. We have now clearly stated the number of events included in the multivariate binary logistic regression model. The model was constructed to identify factors associated with the presence of HCC. Specifically, the multivariate models included a total of 165 HCC cases (events) and 274 CLD controls without HCC. This information has been added to the Results section of the revised manuscript. We added the following statement to the Results section (Page 9, Line, 293-295):
- In the multivariate logistic regression models, the outcome variable was the presence of HCC. The models included 165 HCC cases (events) and 276 CLD controls without HCC.
